# *FakingRecipe*: Detecting Fake News on Short Video Platforms from the Perspective of Creative Process

## ABSTRACT

As short-form video-sharing platforms become a significant channel for news consumption, fake news in short videos has emerged as a serious threat in the online information ecosystem, making developing detection methods for this new scenario an urgent need. Compared with that in text and image formats, fake news on short video platforms contains rich but heterogeneous information in various modalities, posing a challenge to effective feature utilization. Unlike existing works mostly focusing on analyzing *what is presented*, we introduce a novel perspective that considers *how it might be created*. Through the lens of the creative process behind news video production, our empirical analysis uncovers the unique characteristics of fake news videos in material selection and editing. Based on the obtained insights, we design **FakingRecipe**, a creative process-aware model for detecting fake news short videos. It captures the fake news preferences in material selection from sentimental and semantic aspects and considers the traits of material editing from spatial and temporal aspects. To improve evaluation comprehensiveness, we first construct FakeTT, an English dataset for this task, and conduct experiments on both FakeTT and the existing Chinese FakeSV dataset. The results show FakingRecipe's superiority in detecting fake news on short video platforms.

## CCS CONCEPTS

• **Information systems** → **Multimedia information systems**;
• **Security and privacy** → *Human and societal aspects of security and privacy.*

## KEYWORDS

Fake News Video Detection, Multimodal Computing

## 1 INTRODUCTION

In recent years, short-form video-sharing platforms like TikTok have been increasingly popular on mobile Internet and revolutionizing how people consume news [15, 30]. According to Pew Research Center, by 2023, 33% of U.S. adults have ever used TikTok [13], with nearly 43% of these users frequently sourcing their news from this platform [29]. However, the prevalence of news consumption on short video platforms also encourages the emergence and spread

**How Might A Fake News Video Be Created?**

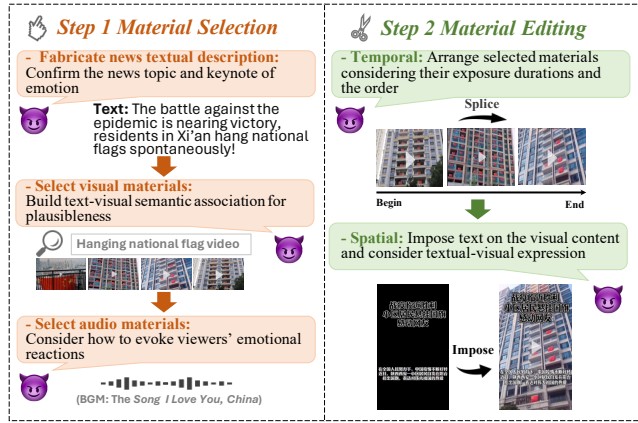

**Figure 1: A fake news video about residents hanging national flags amid the COVID-19 pandemic in China, exhibited along with the speculated creative process. The text was translated into English.**

of fake news videos, posing new serious threats to the online information ecosystem [4, 44]. Consequently, customizing methods for detecting short video fake news is of urgent need.

Unlike fake news in text or image formats, fake news on short video platforms shows unique characteristics and is increasingly indistinguishable from real news, posing new challenges to developing effective detectors [4]. First, the easy-to-use video editing tools largely democratize news creation and enable almost everyone to edit a news video on par with professional journalists [31], making the edit traces widely exist in both real and fake news videos. Second, due to the public nature of short video platforms, even a real news video is likely to be repurposed or re-edited for news faking. However, existing methods for fake news video detection mostly follow ideas from the research line of text-image-based detection and focus on modeling *what is presented* via analyzing the authenticity of multimodal content (*e.g.*, detecting *deepfakes* [12]) and modeling cross-modal correlation [7, 35, 41], which are more likely to be misled by edited and repurposed contents. Faced with the more vague boundary between the real and the fake, it is necessary to find new perspectives and capture more effective clues for fake news video detection.

**In this paper, we propose to switch the perspective from analyzing *what is presented in a fake news video* to considering *how it might be created.*** Our idea is based on a straightforward assumption: Fake news creators on short video platforms often lack first-hand, genuine news materials and professional editing skills while aiming to produce fake news for specific purposes intentionally [42]. This might leave unique characteristics of the resulting

video. Figure 1 provides an intuitive example of the creative process of a fake news video about hanging national flags during the COVID-19 pandemic. The process typically unfolds in two main phases: **material selection** and **material editing**. For selecting the material, the creator first confirmed the news topic (i.e., residents hanging national flags during the pandemic) and the positive sentiment keynote and crafted an attractive narrative that diverges from the truth. Due to the lack of real visual materials (unlike real news), the creator had to repurpose historical materials collected from the Internet to make the fake video more convincing. Finally, an emotionally charged song is selected to impress audiences and achieve its underlying purpose. For the editing phase, the creator might consider arranging materials from the temporal and spatial views with the help of simple editing techniques. Due to the constraint of material sufficiency and editing skills, the collected visual materials were arranged with simple splicing temporally, and the text material was then spatially overlaid on the visual content for a straightforward textual-visual expression. Through this example, we intuitively find that the production of fake news videos may leave the nuances different from that of real ones in terms of material selection and editing. Therefore, modeling from the creative process perspective may help us capture more valuable instrumental clues for fake news video detection.

Inspired by the observation, in this paper, we first quantitatively examine how effective the clues from the creative process perspective are in distinguishing fake and real news videos via an empirical analysis (Section 2). The results validate that statistical discrepancies exist between real and fake news videos in material selection and editing. For instance, we find that compared with real ones, fake news videos exhibit a propensity for selecting more emotionally charged music, using a limited palette of colors, and adopting a less dynamic on-screen text presentation. Based on the empirical analysis, we design **FakingRecipe**, a creative process-aware model for detecting fake news short videos.[1] FakingRecipe is a dual-branch network that models the characteristics of material selection and editing. In the two branches, the Material Selection-Aware Modeling (MSAM) module extracts multimodal features via attention to capture the sentiment resonance between audio and text and the semantic relevance between text and visual frames. The Material Editing-Aware Modeling (MEAM) module models typical spatial and temporal editing behaviors, via 1) analyzing the visual area and on-screen texts for the spatial editing; and 2) building hierarchical temporal structure that considers both intra- and inter-segment fusion for temporal editing. Ultimately, predictions from both branches are integrated through a late fusion function for the final prediction. Experiments on two real-world datasets demonstrate the superiority of the proposed FakingRecipe over seven baseline methods. Our main contributions are as follows:

- **Idea:** We for the first time consider the creative process as a pivotal aspect for detecting fake news on short video platforms and demonstrate the feasibility of this perspective through empirical analysis.
- **Method:** We propose FakingRecipe, a novel dual-branch model for fake news video detection that captures useful clues from

---

[1] The creative process of faking a news video is metaphorically similar to cooking a dish following a recipe, so we use **FakingRecipe** to highlight the model's uniqueness.

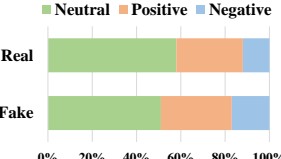

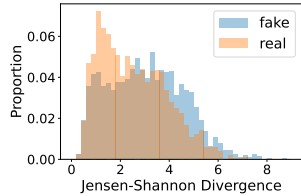

Figure 2: Sentiment analysis of audio material.

Figure 3: JS divergence between textual and visual materials.

the perspective of the creative process, *i.e.*, material selection and editing phases.

- **Resource & Effectiveness:** We construct FakeTT, an English short video dataset for fake news detection. Extensive experiments on both FakeTT and the public Chinese FakeSV dataset show the superiority of FakingRecipe over existing methods in fake news video detection. We will publicly release the new dataset to facilitate further research.

## 2 EMPIRICAL ANALYSIS

We exhibit the manifestation differences between real and fake news videos in different phases of news video creation by conducting empirical analysis on real-world datasets, including the publicly available Chinese dataset FakeSV [35] and the newly constructed English dataset FakeTT. We identify the discrepancies between real and fake news production processes and provide plausible explanations for these phenomena, highlighting the nuances in the creative process to evaluate the news video veracity. Considering that consistent results were observed across both datasets, we only present findings from FakeSV here due to space limitations and attach results on FakeTT in the appendix.

### 2.1 Phase I : Material Selection

**Observation 1.** When selecting audio materials, fake news tends to opt for more emotionally charged audio.

Considering background music (BGM) is a predominant option for short video news creators and the nature of BGM it serves primarily to evoke emotional responses, our analysis of audio selection behaviors focuses on the emotional aspect. We leverage the pre-trained wav2vec model [39] that has been fine-tuned for audio emotion classification. Depicted in Figure 2, we can see that fake news videos exhibit an inclination towards using emotionally charged audio. Given that prior work [9] has indicated emotionality significantly boosts content sharing, we attribute this bias in audio selection to creators' intentions to maximize viewer engagement.

**Observation 2.** When selecting visual materials, fake news often employs clips that exhibit a relatively lower semantic consistency with the accompanying text.

We analyze creators' visual selection behaviors from the perspective of consistency between selected video materials and accompanying text. Specifically, we leverage the pre-trained text-image representation model CLIP [38] to extract textual and visual features. By normalizing these features and calculating the Jensen-Shannon

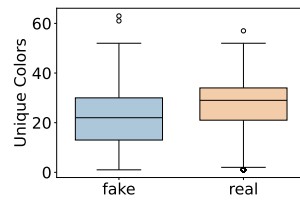

Figure 4: Color richness of on-screen text.

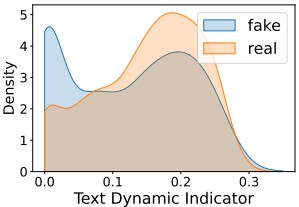

Figure 5: On-screen Text Dynamics.

(JS) Divergence between the attached text and each frame's visual content, we derive an average JS Divergence score across multiple frames as an indicator of text-visual consistency for the entire video. A lower indicator value signifies a higher semantic consistency between the video's textual and visual content. Figure 3 illustrates the distinct distributions of JS Divergence between textual and visual materials for fake and real news. The discrepancies have been statistically confirmed through the Kolmogorov-Smirnov (KS) test, with a p-value of less than 0.05. We find that fake news tends to utilize visual clips with noticeably lower semantic consistency with the accompanying text. We attribute the observed biases in video material selection to the nature that fabricated news inherently lacks access to a rich array of related video materials.

## 2.2 Phase II : Material Editing

We investigate two fundamental editing operations in video creation: spatial editing behaviors and temporal editing behaviors.

**Observation 3.** When spatially imposing text, fake news tends to display relatively plain textual visuals.

Spatial editing behaviors involve overlaying additional layers on top of the original visual materials. This can include adding animated stickers, text, and other elements. Among them, text imposition is a widely used operation in short news videos (85% in the FakeSV dataset), with variations reflected in decisions regarding the text's placement, color, typeface, and font. Here we quantified the color characteristics of the text visual areas in real and fake news videos respectively to explore the differences in color choice behaviors during text imposition. Figure 4 illustrates that real news videos tend to use a richer color palette for text presentation. We attribute this preference to that real news creators often follow conventional editorial norms and invest more effort to improve the presentation quality. Conversely, fake news creators often employ a monochromatic color scheme when imposing text, likely due to a lack of expertise in news production, leaving them unaware of the potential impact these details can have on viewers.

**Observation 4.** When temporally splicing materials, fake news tends to adopt a relatively simple arrangement.

Temporal editing behaviors, on the other hand, refer to the reorganization and splicing of multiple material segments. The duration and positioning of different segments can subtly influence viewers' perceptions of the news video. Here we examine the temporal editing behaviors related to text exposure, analyzing differences in the temporal arrangement of text segments between real and fake

news. Specifically, we developed an indicator, $I_D$, to measure the dynamism of text presentation. By calculating the mean ($\mu$) and standard deviation ($\sigma$) of exposure durations ($d_1, d_2, ...$) for different text phrases within a video, $I_D$ is defined as $\sigma(1 - \mu)$, based on the principle that shorter exposure times and greater variance among text exposure durations indicate stronger text temporal editing dynamism. Figure 5 shows the fitted sample density distribution of the on-screen text dynamic scores in the FakeSV dataset, revealing significant differences between the temporal editing behaviors of real and fake news, with real news exhibiting more dynamic text presentations. We ascribe this tendency to two factors: First, the disparity in video creation capabilities, wherein most creators of authentic news, endowed with professional media training, possess a nuanced understanding of effectively integrating text with visual elements. Second, the constraints posed by the availability of materials. Fabricated news, inherently characterized by its scant and biased content, often lacks the robust information necessary for dynamic presentations. This deficiency compels creators to resort to the static placement of limited information in specific areas of the screen.

## 3 METHOD

### 3.1 Overview

Drawing on the insights from our empirical analysis, we present FakingRecipe (Figure 6), a creative process-aware fake news video detection model. The model observes the given news video from the two pivotal phases of the creative process to unearth veracity indicating clues. Treating the feature from two phases as independent viewpoints, FakingRecipe is structured with dual branches operating separately and employs a late fusion strategy to integrate predictions from these independent perspectives.

### 3.2 Material Selection-Aware Modeling (MSAM)

Based on prior analysis, we examine the creators' material selection behavior from two aspects (*i.e.*, sentiment and semantic). The dominant role of different modalities varies in conveying information: Audio primarily expresses emotion, text renders emotional tones while conveying semantic information, and visuals generally complement the text to communicate semantic content. Therefore, we strategically select combinations of modalities for multifaceted feature extraction, subsequently fusing multimodal features from multiple viewpoints.

Specifically, for the sentimental aspect, we consider audio and textual content as the primary sources. We utilize fine-tuned versions of HuBERT [17] and XLM-RoBERTA [8] as encoders to extract audio sentimental features $\mathbf{H}_{SEN-A}$ and textual sentimental features $\mathbf{H}_{SEN-T}$, respectively. These sentimental features from different modalities are then concatenated and fed into a standard Transformer layer [46]. By leveraging self-attention, the transformer layer fuses multimodal sentimental features to produce a unified sentimental feature representation $\mathbf{H}_{SEN}$.

In the semantic aspect, visual and textual contents take precedence, while the audio mainly serves as background music, playing a minimal role. Keyframes are extracted from videos, serving as the basis for visual analysis. Utilizing CLIP [38], we encode text and keyframes to token/frame-level text semantic features $\mathbf{H}_{SEM-T}$ and

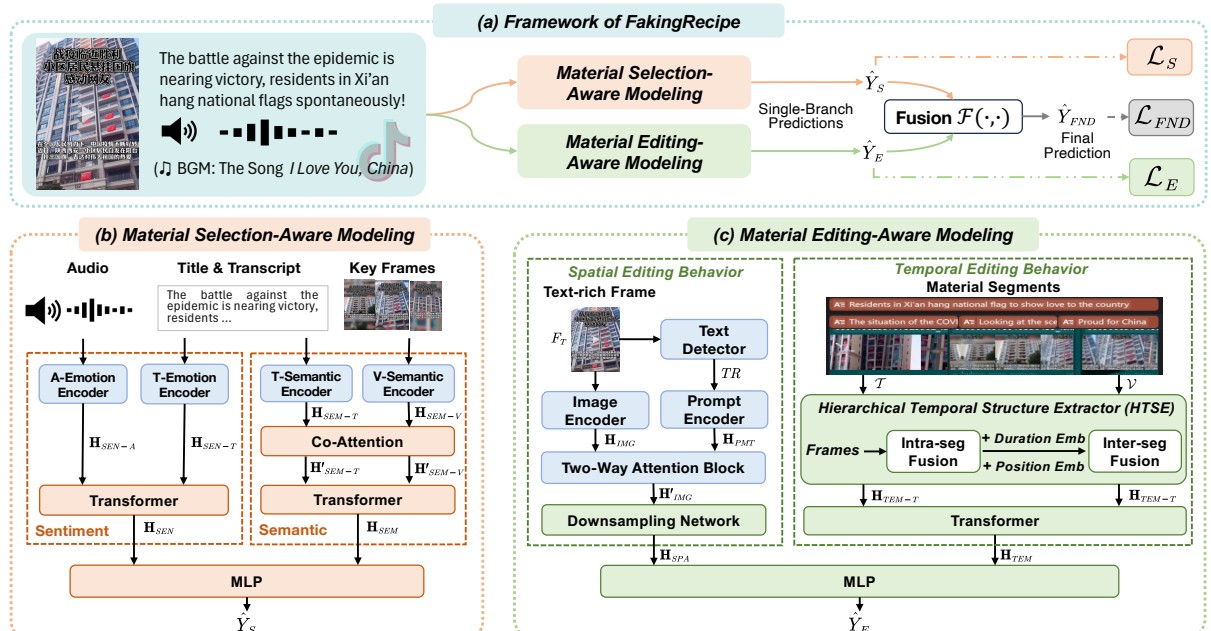

Figure 6: Overview of the proposed FakingRecipe model. (a) Overall framework: The news video is processed through dual perspectives, with a late fusion strategy employed to integrate clues for final prediction. (b) Material Selection-Aware Modeling (MSAM) module: Extracts clues from both sentimental and semantic aspects. (c) Material Editing-Aware Modeling (MEAM) module: Extracts clues based on spatial and temporal aspects. $\mathcal{F}(\cdot, \cdot)$ denotes the fusion function. The parameters in the modules in blue are frozen and others are trainable. The overall model is trained under the supervision of the loss functions $\mathcal{L}_{FND}$, $\mathcal{L}_S$, and $\mathcal{L}_E$. The text in this case is translated into English.

visual semantic features $\mathbf{H}_{SEM-V}$. Interaction between text and visual content is facilitated through a co-attention transformer [27], resulting in visually enhanced textual features $\mathbf{H'}_{SEM-T}$ and textually enhanced visual features $\mathbf{H'}_{SEM-V}$. These features are then averaged, concatenated, and input into a transformer layer mirroring the structure used in the sentimental analysis. This process integrates semantic features from various modalities into a singular semantic feature representation $\mathbf{H}_{SEM}$.

The sentimental feature $\mathbf{H}_{SEN}$ and semantic feature $\mathbf{H}_{SEN}$ are then concatenated and fed into a two-layer MLP to derive the fake news predicted score $\hat{Y}_S$ from the material selection analysis perspective:

$$\hat{Y}_S = \text{MLP}([\mathbf{H}_{SEN}; \mathbf{H}_{SEM}]). \quad (1)$$

## 3.3 Material Editing-Aware Modeling (MEAM)

In mining detecting clues from the perspective of creator editing behaviors, we focus on spatial and temporal aspects, identified as critical in our empirical analysis.

**Spatially**, we examine the prevalent practice of imposing text. Given a video $V$, we select a representative text-rich frame $F_T$, identified based on the size of the text presence area, as our starting point. We first employ an OCR spotting model, CRAFT [1], to delineate text regions $TR = \{\text{box}_1, \text{box}_2, ...\}$ within $F_T$. These regions are subsequently transformed into prompt embeddings $\mathbf{H}_{PMT}$ employing a methodology inspired by the prompt encoder in Segment Anything Model (SAM) [23]. In parallel, $F_T$ undergoes processing

by a pre-trained Vision Transformer (ViT) [10] to produce initial encodings $\mathbf{H}_{IMG}$. Both $\mathbf{H}_{IMG}$ and $\mathbf{H}_{PMT}$ are then fed into a Two-Way Attention block, mirroring the structure used by the SAM's mask decoder. The block leverages both prompt self-attention and cross-attention mechanisms, functioning in two directions (prompt-to-image and image-to-prompt). This dual attention strategy is employed to refine the initial visual encoding $\mathbf{H}_{IMG}$, ensuring it focuses more accurately on text regions within the frame. Following attention processing, the updated $\mathbf{H'}_{IMG}$ undergoes downsampling via two convolutional layers and then flattened to derive the spatial pattern feature $\mathbf{H}_{SPA}$. Figure 7 provides a detailed depiction of the Two-Way Attention block and the downsampling network.

**Temporally**, we examine the splicing practice of text segment and video segment. The audio track is omitted due to the observation that most audio tracks consist of continuous background music. For a Video $V$, preprocessing extracts a sequence of text content $\mathcal{T} = \{(t_1, d_1), (t_2, d_2), ...(t_n, d_n)\}$ and a sequence of visual content $\mathcal{V} = \{(v_1, d_1), (v_2, d_2), ...(v_m, d_m)\}$, with $n$ and $m$ indicating the counts of text and video segments respectively. $t_i$ represents the $i$-th textual segment, $v_i$ denotes the middle frame of the $i$-th video clip, and $d_i = [\text{FrameIdx}_i^{begin}, \text{FrameIdx}_i^{end}]$ marks the time interval of the $i$-th segment's appearance. The input also incorporates frame rate (fps) and the total frame count (vframes) of the video to contextualize duration. Each modality's temporal structure is initially modeled separately, followed by an interaction phase to derive overall temporal editing features. Specifically, we design a

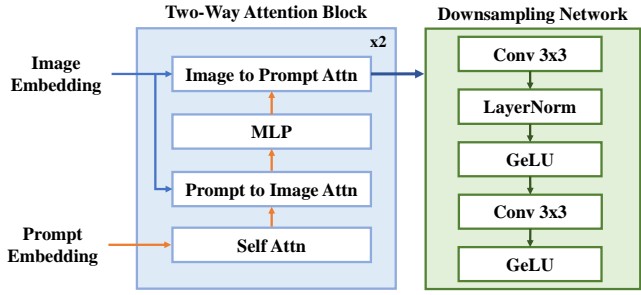

**Figure 7: Details of the Two-Way Attention Block and the Downsampling Network.**

Hierarchical Temporal Structure Extractor (HTSE) for extracting temporal structure features applicable to both modalities. HTSE first performs intra-segment fusion for content occurring in the same time span to derive segment content features $\mathbf{Seg}_i$. For text, $\mathbf{Seg}_i^T$ is obtained by concatenating multiple segments and encoding them collectively, while for visuals, it applies the self-attention ($SA$) mechanism for integration:

$$\mathbf{Seg}_i^V = \text{MEAN}(\text{SA}([v_1, v_2, ..., v_k])), \tag{2}$$

where k is the frame count within segment $i$, and MEAN($\cdot$) denotes the mean pooling. To model the subtle influences of the duration and temporal position of different segments, we introduce each segment's temporal position and exposure duration information. Positional encoding (**PE**) is generated using sine and cosine functions to reflect each segment's temporal position, akin to that leveraged by Transformer [46]:

$$\text{PE}_i^{(ei)} = \begin{cases} \sin\left(w_k i\right), & \text{if } ei = 2k \\ \cos\left(w_k i\right), & \text{if } ei = 2k+1, \end{cases} \tag{3}$$

where $w_k = 1/(10000^{2k/\dim_{PE}})$ represents the frequency of the sinusoid for each dimension and $\mathbf{PE}_i$ is the $i$-th segment's positional embedding. For duration encoding (**DE**), we employ an equi-frequency binning approach, determining duration groups through empirical analysis and assigning a learnable embedding to each group. Considering the audience's perception of exposure time in reality, both absolute and relative durations are evaluated:

$$\text{Dura}_i^{abs} = (\text{FrameIdx}_i^{begin} - \text{FrameIdx}_i^{end})/\text{fps},$$

$$\text{Dura}_i^{rel} = (\text{FrameIdx}_i^{begin} - \text{FrameIdx}_i^{end})/\text{vframes}.$$

Here $\text{Dura}_i^{abs}$ and $\text{Dura}_i^{rel}$ denote the absolute (in seconds) and relative (the proportion of the total duration) durations of segment $i$, respectively. The $i$-th segment's duration embedding is:

$$\mathbf{DE}_i = [\text{Emb}(\text{Group}(\text{Dura}_i^{abs})); \text{Emb}(\text{Group}(\text{Dura}_i^{rel}))], \quad (4)$$

where Group($\cdot$) maps a duration to its designated group and Emb($\cdot$) retrieves the corresponding embedding for that group.

Integrating positional and duration encodings, the segment features are updated to $\mathbf{SEG}_i$, serving as the input for inter-segment fusion, which captures the relationships between different segments

**Table 1: Statistics of two datasets for evaluation.**

| Dataset | Time Range | Avg Duration (s) | #Fake | #Real | #All |
|---|---|---|---|---|---|
| FakeSV | 2017/10-2022/02 | 39.88 | 1,810 | 1,814 | 3,624 |
| FakeTT | 2019/05-2024/03 | 47.69 | 1,172 | 819 | 1,991 |

using a similar self-attention mechanism, generating temporal pattern features for each modality:

$$\mathbf{SEG}_i = \mathbf{Seg}_i + \mathbf{PE}_i + \mathbf{DE}_i, \tag{5}$$

$$\mathbf{H}_{TEM-M} = \text{MEAN}(\text{SA}([\mathbf{SEG}_1^M, \mathbf{SEG}_2^M, ...])). \tag{6}$$

Utilizing HTSE, we derive temporal editing features for both text ($\mathbf{H}_{TEM-T}$) and visual ($\mathbf{H}_{TEM-V}$) modalities, and they are subsequently processed by a standard Transformer layer to produce the consolidated temporal editing feature $\mathbf{H}_{TEM}$. The spatial editing feature $\mathbf{H}_{SPA}$ and the temporal editing feature $\mathbf{H}_{TEM}$ are then concatenated and fed into a two-layer MLP to compute the fake news predicted score $\hat{Y}_E$ from the material editing analysis perspective:

$$\hat{Y}_E = \text{MLP}([\mathbf{H}_{SPA}; \mathbf{H}_{TEM}]. \tag{7}$$

## 3.4 Predication and Optimization

*3.4.1 Prediction.* Building on the predicted scores $\hat{Y}_S$ from material selection modeling and $\hat{Y}_E$ from material editing modeling, we adopt a late fusion strategy to get the final score $\hat{Y}_{FND}$:

$$\hat{Y}_{FND} = \mathcal{F}(\hat{Y}_S, \hat{Y}_E) = \hat{Y}_S * \tanh(\hat{Y}_E), \tag{8}$$

where $\mathcal{F}(\cdot, \cdot)$ is the fusion function. Inspired by previous works [6, 49], we adopt the tanh($\cdot$) function, which introduces non-linearity to enhance the fusion strategy's representational capacity.

*3.4.2 Optimization.* Following previous works [7, 35, 41], we utilize cross-entropy loss to optimize our model:

$$\mathcal{L}_{FND} = \text{Cross-Entropy}(\hat{Y}_{FND}, Y), \tag{9}$$

where $Y$ is the ground-truth label for each short video news.

To further supervise the material selection and material editing modeling, the final loss $\mathcal{L}$ incorporates the loss for $\hat{Y}_S$ and $\hat{Y}_E$:

$$\mathcal{L} = \mathcal{L}_{FND} + \alpha\mathcal{L}_S + \beta\mathcal{L}_E, \tag{10}$$

where $\alpha$ and $\beta$ are hyperparameters that balance the impacts on the back-propagation of the three branches. $\mathcal{L}_S$ and $\mathcal{L}_E$ denote the Cross-Entropy($\hat{Y}_S, Y$) and Cross-Entropy($\hat{Y}_E, Y$), respectively.

## 4 EXPERIMENTS

In this section, we conduct extensive experiments on two real-world datasets to verify the effectiveness of FakingRecipe by comparing it with seven representative baselines and the FakingRecipe variants.

## 4.1 Datasets

To validate the generalizability of the proposed FakingRecipe, we conduct experiments on two datasets of different languages:

**FakeSV**[2]: The largest publicly available Chinese dataset for fake news detection on short video platforms, featuring samples from *Douyin* and *Kuaishou*, two popular Chinese short video platforms. Each sample in FakeSV contains the video itself, its title, comments,

---

[2]https://github.com/ICTMCG/FakeSV

Table 2: Performance comparison between FakingRecipe and baselines on the FakeSV and FakeTT datasets. The best performance in each column is bolded and the relative improvement of FakingRecipe over the best baseline is in the brackets.

| Dataset | Method | Accuracy | Macro F1 | Fake | | | Real | | |
|---|---|---|---|---|---|---|---|---|---|
| | | | | Precision | Recall | F1 | Precision | Recall | F1 |
| FakeSV | GPT-4 | 67.43 | 67.34 | 83.71 | 53.99 | 65.64 | 57.81 | **85.71** | 69.05 |
| | GPT-4V | 69.15 | 69.14 | 82.35 | 58.78 | 68.60 | 60.00 | 83.08 | 69.68 |
| | HCFC-Hou | 74.91 | 73.61 | 73.46 | 86.51 | 79.46 | 77.72 | 60.08 | 67.77 |
| | HCFC-Medina | 76.38 | 75.83 | 77.50 | 81.58 | 79.49 | 74.77 | 69.75 | 72.17 |
| | FANVM | 79.52 | 78.81 | 78.64 | 87.17 | 82.68 | 80.98 | 69.75 | 74.94 |
| | TikTec | 73.43 | 73.26 | 78.37 | 72.70 | 75.43 | 68.08 | 74.37 | 71.08 |
| | SVFEND | 80.88 | 80.54 | **85.82** | 77.63 | 81.52 | 74.53 | 83.61 | 78.81 |
| | **FakingRecipe (Ours)** | **85.35**$_{(+5.53\%)}$ | **84.83**$_{(+5.33\%)}$ | 83.33 | **92.11** | **87.50** | **88.35** | 76.47 | **81.98** |
| FakeTT | GPT-4 | 61.45 | 60.66 | 43.36 | 75.61 | 55.11 | 83.19 | 55.00 | 66.22 |
| | GPT-4V | 58.69 | 58.69 | 44.52 | **88.46** | 59.23 | 88.00 | 43.42 | 58.15 |
| | HCFC-Hou | 73.24 | 72.00 | 56.93 | 78.79 | 66.10 | 87.04 | 70.50 | 77.90 |
| | HCFC-Medina | 62.54 | 62.23 | 46.24 | 80.81 | 58.82 | 84.92 | 53.50 | 65.64 |
| | FANVM | 71.57 | 70.21 | 55.15 | 75.76 | 63.83 | 85.28 | 69.50 | 76.58 |
| | TikTec | 66.22 | 65.08 | 49.32 | 72.73 | 58.78 | 82.35 | 63.00 | 71.39 |
| | SVFEND | 77.14 | 75.63 | 62.33 | 78.79 | 69.57 | 87.91 | 76.33 | 81.69 |
| | **FakingRecipe (Ours)** | **79.15**$_{(+2.61\%)}$ | **77.74**$_{(+2.79\%)}$ | **64.75** | 81.82 | **72.18** | **89.74** | **77.83** | **83.30** |

metadata, and publisher profiles. We do not use the last three values to focus on understanding the content itself.

**FakeTT**: Our newly constructed English dataset for a comprehensive evaluation in English-speaking contexts.[3] Curated from TikTok, this dataset follows a similar collection process to [35], focusing on videos related to events reported by the fact-checking website Snopes[4]. Each video was rigorously annotated for authenticity by at least two independent annotators, resulting in a collection of 1,172 fake news videos and 819 real news ones from May 2019 to March 2024, with video, audio, and text description (title) available. See more details in the appendix.

Table 1 shows the statistics of the two datasets. To simulate real-world scenarios, we adopt a temporal split strategy for our experiments, dividing the data chronologically into training, validation, and testing sets with ratios of 70%, 15%, and 15%, respectively. Such a data split reflects the potential of applying compared methods in reality.

## 4.2 Experimental Setup

*4.2.1 Baselines.* We compare the proposed FakingRecipe with a range of state-of-the-art baselines, including handcrafted features-based baselines, neural networks-based baselines, and (multimodal) large language model ((M)LLMs) baselines:

**Handcraft Feature-based Baselines:** (1) **HCFC-Hou [16]** utilizes linguistic features from speech, acoustic emotion features, and user engagement statistics with a linear kernel SVM for classification. (2) **HCFC-Medina [40]** extracts TF-IDF vectors from video titles and the first hundred comments, applying SVM for detection.

**Neural Network-based Baselines:** (1) **FANVM [7]** harnesses visual features from keyframes and textual features from titles and comments, using an adversarial network to extract topic-agnostic multimodal features for classification. (2) **TikTec [41]** employs

---

[3]Shang et al. [41] did collect an English TikTok dataset but did not release it. We did not receive any reply to our email for the dataset inquiry.
[4]https://www.snopes.com/

speech text-guided visual object features and MFCC-guided speech textual features, using a co-attention mechanism for fusion and classification. (3) **SVFEND [35]** leverages cross-modal transformers to boost interaction between modalities and integrates content with social context features via a self-attention mechanism.

**(M)LLM Baselines:** (1) **GPT-4 [32]** is one of the most powerful LLMs currently and is used to make predictions based on video news titles and extracted on-screen text. We use a zero-shot prompt template inspired by Hu et al. [18]. (2) **GPT-4V [52]** is the variant of GPT-4 that supports visual inputs. We include the video's thumbnail in the inputs to explore the capabilities of (M)LLMs in this task.

Given that we focus on content-only detection at the early news spreading stage, all baselines are adapted to rely solely on content.

*4.2.2 Implementation Details.* For data preprocessing, we select the frame with the largest text region for spatial editing feature learning and segment video frames using TransNetv2 [43] for temporal behavior modeling. All the MLPs in our experiments consist of three layers with a ReLU activation and a dropout rate of 0.1. The co-attention mechanism features four heads, and convolutional layers in the downsampling network are configured with a stride of 2 and padding of 1. Training parameters include setting hyperparameters $\alpha$ and $\beta$ at 0.1 and 2.0, respectively, learning rates of 5e-5 for FakeSV and 1e-3 for FakeTT, and a batch size of 128. The model undergoes training for 30 epochs, incorporating early stopping to mitigate overfitting, and employs the Adam [22] for optimization. We report the average results of multiple runs.

*4.2.3 Metrics.* We report Accuracy and macro F1 as primary evaluation metrics, which are widely used in existing works [35, 37]. To account for imbalanced label distributions, we additionally report the F1-score, Precision, and Recall for each label (i.e., Fake or Real).

## 4.3 Overall Performance

Table 2 presents the performance of FakingRecipe and the compared baselines. The results reveal several key observations:

**Table 3: Ablation study of multiple model components.**

| MSAM | | MEAM | | FakeSV | | FakeTT | |
|---|---|---|---|---|---|---|---|
| SEN | SEM | SPA | TEM | Acc | F1 | Acc | F1 |
| ✓ | ✓ | ✓ | ✓ | **85.35** | **84.83** | **79.15** | **77.74** |
| ✓ | ✓ | | | 83.94 | 83.56 | 77.92 | 76.61 |
| | | ✓ | ✓ | 82.47 | 81.96 | 71.24 | 69.81 |
| | ✓ | ✓ | ✓ | 83.58 | 83.10 | 76.92 | 75.95 |
| ✓ | | ✓ | ✓ | 84.31 | 83.92 | 74.91 | 73.71 |
| ✓ | ✓ | | ✓ | 84.87 | 84.42 | 78.76 | 77.39 |
| ✓ | ✓ | ✓ | | 84.14 | 83.81 | 78.59 | 77.53 |

(Module / Dataset header spanning above)

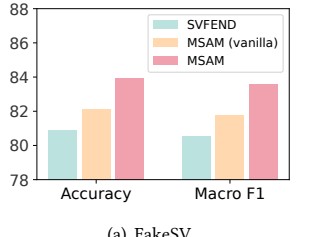
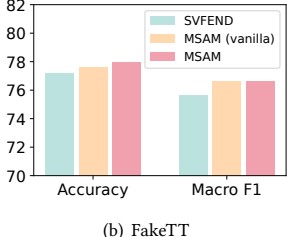

(a) FakeSV        (b) FakeTT

**Figure 8: Performance comparison of the proposed MSAM module and the best baseline SVFEND on the FakeSV and FakeTT datasets.**

First, the zero-shot (M)LLM-based methods, namely GPT-4(V), underperform the methods specifically tailored for fake news video detection, which indicates the complexity of the task and the necessity of specialized models for this task currently. Notably, GPT-4(V) exhibits biases in authenticity judgments, tending to classify videos as real in the FakeSV dataset and as fake in the FakeTT dataset, possibly due to different knowledge accumulation in the tested large models.

Second, neural network-based baselines generally outperform handcraft feature-based baselines, demonstrating the superiority of automated neural network models in handling complex fake news detection tasks. However, in some instances, the handcraft feature-based baselines surpass certain neural network models, particularly TikTec. This suggests that integrating human-guided knowledge into the models may bring additional advantages in specific cases.

Finally, FakingRecipe outperforms all competing methods in Accuracy and macro F1 on both datasets, validating its effectiveness in detecting fake news videos. Notably, the improvements are more pronounced on the FakeSV dataset (5.53% increase in accuracy and 5.33% in macro F1) compared to that on FakeTT (2.61% in accuracy and 2.79% in macro F1), possibly reflecting the moderate pattern differences of the creative process in different cultural background.

## 4.4 Ablation Study

To rigorously evaluate the individual contributions of each component within FakingRecipe, we conduct extensive ablation studies, the results of which are detailed in Table 3. We first focus on the performances of the two core modules: Material Selection-Aware Modeling (MSAM) and Material Editing-Aware Modeling (MEAM). It is observed that MSAM generally outperforms MEAM, with a more notable performance gap observed on the FakeTT dataset compared to FakeSV. While MEAM showed relatively lower performance on its own, it provides crucial complementary insights that significantly enhance the overall effectiveness of the combined model beyond what is achieved by MSAM alone.

Further exploration into each specific aspect within these modules: sentimental and semantic for MSAM, and spatial and temporal for MEAM. By systematically removing each aspect and comparing the altered model's performance to the original, the results confirm that each component plays a vital role in the model's overall effectiveness. Among these aspects, while the spatial component shows the smallest improvement in performance, the sentimental aspect is most impactful for FakeSV, and the semantic aspect is particularly effective for FakeTT. This detailed analysis not only demonstrates

the essential contribution of each aspect but also underscores the synergy that their integration brings to the effectiveness of FakingRecipe in detecting fake news videos.

## 4.5 Further Analysis

The performance improvements in FakingRecipe are attributed to the enhancements by the MSAM and MEAM modules. Specifically, MSAM facilitates multimodal content understanding and MEAM introduces a novel perspective on mining content utilization. In this section, we conduct a deeper investigation into these two modules and present two additional findings:

**The synergy of creative process-aware encoding and fusion strategy deepens understanding of video materials, leading to better detection performance.** We implement a simplified version of MSAM, termed MSAM (vanilla), which directly concatenates features from multiple encoders for classification. As depicted in Figure 8, MSAM (vanilla) performs better than SVFEND which employs universal encoders for multimodal content understanding, confirming the efficacy of our material selection-aware multimodal content encoding strategy. However, its performance still falls behind the full MSAM configuration, emphasizing the crucial role of the advanced fusion structure in performance improvement. This exploration underscores the individual effectiveness of both the encoding and fusion strategy and their synergy within MSAM.[5]

**Creative process-aware modeling introduces new effective clues that can even bring improvements to other existing models.** We assess the generalizability of MEAM, which introduces a novel perspective in modeling multimodal content utilization. We directly integrate MEAM into TikTec and SVFEND using the same late fusion strategy as FakingRecipe. The results on two datasets are shown in Table 4. We see that incorporating MEAM resulted in performance gains on both baselines, with TikTec showing significant improvements, affirming MEAM's capacity to elevate performance under effective fusion.

## 4.6 Case Study

We further demonstrate the complementary capabilities of MSAM and MEAM in detecting fake news videos through two real examples from the FakeSV dataset in Figure 9. In the left example, a video with high-quality production and visually rich materials is evaluated. Influenced by the video's polished appearance, MEAM

---

[5] The analysis of different fusion strategies is in the appendix.

**Table 4: Performance comparison of different models enhanced by our proposed MEAN module on two datasets.**

| Method | FakeSV | | FakeTT | |
|--------|----------|----------|----------|----------|
| | Accuracy | Macro F1 | Accuracy | Macro F1 |
| TikTec | 73.43 | 73.26 | 66.22 | 65.08 |
| (+MEAM) | 83.95 | 83.52 | 71.57 | 70.61 |
| SVFEND | 80.88 | 80.54 | 77.14 | 75.63 |
| (+MEAM) | 83.03 | 82.37 | 78.76 | 77.15 |

classifies it as real. However, MSAM assesses the situation from a different angle, detecting emotionally charged language in the video's title, such as "what a heinous act," which identifies as a potential indicator of misinformation. This nuanced analysis by MSAM accurately flags the video as fake, showcasing its ability to probe deeper than superficial qualities. Conversely, the right example presents a video with a neutral expression, which initially leads MSAM to classify it as authentic. Here, MEAM provides critical complementary information. It scrutinizes the video's sparse visual content and simplistic textual presentation, cues that suggest a lack of authenticity. This focused evaluation by MEAM correctly identifies the video as fake, highlighting its essential role in the overall analysis. These case studies underscore the complementary nature of MSAM and MEAM in FakingRecipe, enabling a layered and comprehensive assessment of news videos. We provide the failure case analysis in the appendix.

## 5 RELATED WORK

**Fake News Video Detection.** The early work closely related to fake news video detection traces its roots to multimedia forensics research. Forensics-based works follow a basic idea about veracity that misinformation videos are often produced using forgery techniques [4, 12]. However, with the prevalence of user-friendly editing tools, manipulating visual content has become a common practice across social media platforms, significantly limiting the applicability of this detection approach. Thus, recent investigations have shifted their methodology towards mining detection clues from multimodal content. Handcraft features tailored for fake news video detection [16, 33, 34, 40] like linguistic patterns, acoustic emotion, and user engagement statistics are designed. Further studies incorporate visual expression and leverage deep neural networks [7, 19, 24, 26, 35, 41] for falsehood identification. Building on the foundation of multimodal content clues within individual samples, some researchers propose to incorporate the neighborhood relationship in an event for fake news video detection, exemplified by the NEED framework [37]. Though effective, its dependency on existing data accumulations limits its applicability in real-world scenarios. Instead, our method is suitable for detection at the early stage as it only requires the video content as the input.

**Narrative-aware Fake News Detection.** News reporting has long been seen as involving a form of storytelling [2, 45, 47]. From this perspective, applying narrative theory, a discipline focusing on how stories are depicted persuasively [3], to characterize fake news emerges as an intuitive idea. Narrative theory emphasizes analyzing the "what" (the content of the story) and the "how" (the strategy of storytelling) as its two pivotal aspects [11], echoing the perspective of the creative process. The potential of applying

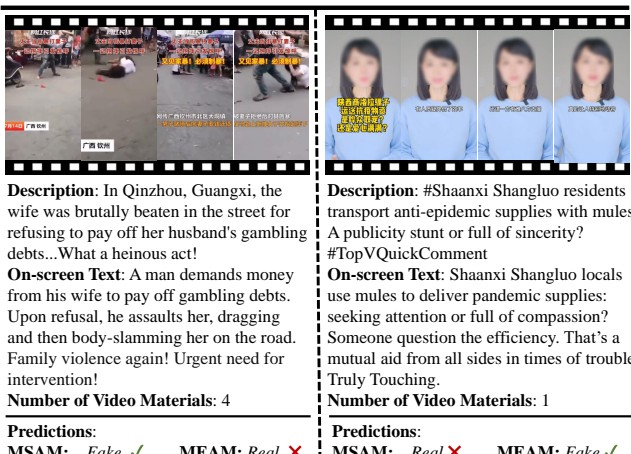

**Description**: In Qinzhou, Guangxi, the wife was brutally beaten in the street for refusing to pay off her husband's gambling debts...What a heinous act!
**On-screen Text**: A man demands money from his wife to pay off gambling debts. Upon refusal, he assaults her, dragging and then body-slamming her on the road. Family violence again! Urgent need for intervention!
**Number of Video Materials**: 4

**Predictions**:
**MSAM**: *Fake* ✓    **MEAM**: *Real* ✗
**FakingRecipe**: *Fake* ✓

**Description**: #Shaanxi Shangluo residents transport anti-epidemic supplies with mules: A publicity stunt or full of sincerity? #TopVQuickComment
**On-screen Text**: Shaanxi Shangluo locals use mules to deliver pandemic supplies: seeking attention or full of compassion? Someone question the efficiency. That's a mutual aid from all sides in times of trouble. Truly Touching.
**Number of Video Materials**: 1

**Predictions**:
**MSAM**: *Real* ✗    **MEAM**: *Fake* ✓
**FakingRecipe**: *Fake* ✓

**Figure 9: Two fake news cases from FakeSV demonstrating the complementary roles of MSAM and MEAM in FakingRecipe. We translate the texts into English and blur the faces to respect user privacy.**

narrative theory for detecting fake news has been demonstrated by studies on news articles [14, 20, 48]. However, within the realm of multimodal news, related research remains limited. Current studies in multimodal fake news detection [5, 21, 36, 50, 51, 53] typically concentrate solely on the analysis of presented content, neglecting the broader narrative structures. Tseng et al. [45] make the first foray into understanding narratives of disinformation in TV news videos. A multimodal discourse analysis scheme is proposed to uncover narrative strategies [2]. However, their focus is to assist manual statistical analysis using web-based tools [25] and thus is inapplicable to automatic detection. Our study takes the first step to detect fake news on short video platforms from the perspective of the creative process, which can be seen as a practical solution of the narrative theory for this task.

## 6 CONCLUSION

We proposed to detect fake news on short video platforms from the perspective of the creative process and designed the creative process-aware detector, FakingRecipe. It observes the given video from material selection and editing perspectives to capture the unique production characteristics of fake news videos. We conducted experiments on the English FakeTT dataset newly constructed by us and the popular Chinese FakeSV dataset and validated the effectiveness of FakingRecipe.

**Limitations and Future Work.** Though bringing a new perspective and experimentally shown effective, our model design mainly relies on empirical analysis, and thus may not fully correspond to the existing theoretical knowledge in the analysis of fake news creation. Since spreading and combating fake news is constantly adversarial, the model may require periodic updates in real applications. In the future, we plan to draw inspiration from journalism and communication literature to make the creative process modeling more intrinsic. Also, it is still worthwhile exploring how to equip (M)LLMs with our method, possibly via advanced techniques [28].

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
