# OpenReview forum: "FakingRecipe: Detecting Fake News on Short Video Platforms from the Perspective of Creative Process"
_acmmm.org/ACMMM/2024/Conference — MM2024 Poster_

### Official Review · Reviewer_ghbw · 2024-05-10

**Rating:** 5
**Confidence:** 3

**Summary:**

The authors propose a novel and interesting perspective and design a creative process-aware model for multimodal fake news detection on short video platforms. Meanwhile, they construct an English dataset Fake TT to evaluate the effectiveness of their methods.

**Strengths:**

1. The objective of this work is clearly stated and paper is well structured.
2. The idea of considering how fake video news might be created is interesting. This novel perspective has the potential to significantly contribute to the advancement of research in this field.
3. Extensive experiments demonstrate the effectiveness of their method.
4. It is commendable to publicly release the new dataset to facilitate further research.

**Limitations:**

1. Due to the limited informational value density of short video content, it seems more reasonable to conduct detection from the perspective of 'how it might be created' than 'what is presented'. However, in real-world settings, the creative patterns of fake news can exhibit considerable variability. Can a creative process-aware model effectively address the complexities of such environments? While this poses a challenging problem, I am curious to know if the author has taken this aspect into consideration.
2. It appears that there are some typos, such as 'ei' in Eq.3.

**Suitability:**

3

---

### Official Review · Reviewer_TYrJ · 2024-05-24

**Rating:** 4
**Confidence:** 3

**Summary:**

The authors propose a novel model that examines the creative process during the production stage of news videos. They introduce FakingRecipe, a model that detects fake news by analyzing the choices in material selection and editing. Besides, the authors construct a new English fake news video detection dataset.

**Strengths:**

1. The authors construct an English fake news video dataset, which is meaningful for the community.
2. The proposed FakingRecipe provides a new respective from the video creator and achieves excellent performance in two datasets.
3. The experiment results are convincing and solid.

**Limitations:**

1. The proposed dataset FakeTT is not open source. The demo or download link should be available.
2. A more detailed collection process should be explained. Besides more statics should be displayed about the dataset.
3. The absence of parameter sensitivity analysis in the experiment.

**Suitability:**

3

---

### Official Review · Reviewer_3QhL · 2024-05-27

**Rating:** 3
**Confidence:** 3

**Summary:**

The paper studies fake news detection on short video platforms from an interesting perspective that considers how it might be created and studies the unique characteristics of fake news videos in material selection and editing empirically. Based on the analysis, the authors propose a FakingRecipe that models material selection from sentimental and semantic aspects and considers the traits of material editing from spatial and temporal aspects. Extensive experiments show the proposed method performs better in detecting fake news on short video platforms.

**Strengths:**

1. Studying how fake news might be created is an interesting perspective. The empirical analysis provides novel insights into the fake news data.
2. The paper is well-written and organized. The proposed method is clearly formatted and easy to follow.
3. The authors deliver an English dataset for fake news detection in the short video platform, contributing to the community.

**Limitations:**

1. The proposed method lacks novelty and hardly corresponds to modeling material selection and editing.
2. The loss functions of the proposed method are confusing. Both the outputs of the two branches and the final outputs are aligned to the ground-truth label Y.
3. The case study is not insightful. The visualized cases fail to reflect the working mechanism and the effects of the two branches of the proposed model.

**Suitability:**

3

---

### Official Review · Reviewer_f3S8 · 2024-05-28

**Rating:** 5
**Confidence:** 3

**Summary:**

The paper introduces an innovative approach for detecting fake news on short video platforms by focusing on the creative process involved in producing these videos. The authors propose the "FakingRecipe," a detection model that evaluates videos from the perspectives of material selection and editing, aiming to identify the unique production characteristics associated with fake news.

**Strengths:**

1. The paper identifies discrepancies between real and fake news production processes and provides plausible explanations for these phenomena, highlighting the nuances in the creative process to evaluate the veracity of news videos.
2. The paper constructs FakeTT, an English short video dataset for fake news detection.
3. The paper proposes FakingRecipe, a novel dual-branch model for fake news video detection that captures useful clues from the perspective of the creative process, i.e., material selection and editing phases.
4. The proposed network achieves state-of-the-art performance on two public real-world datasets.

**Limitations:**

1. Section 3.3 on the spatial aspects lacks some formal mathematical expressions.
2. The implementation details of the Two-Way Attention Block are not provided, specifically the internal details of Image to Prompt Attention and Prompt to Image Attention.
3. Parameter analysis is missing.
4. Implementation details for Table 4 are not provided.

**Suitability:**

3

---

### Meta-Review · Area_Chair_GZJ7 · 2024-06-28

**Recommendation:** Accept (Poster)
**Confidence:** 4

**Metareview:**

The reviewers are unanimous in their opinion that the paper should be accepted to the program of ACM Multimedia 2024. They are particularly positive about the idea of studying the properties of fake news production process. In addition, they appreciated the introduced FakeTT dataset, hoping that it will be made public. However, some reviewers were questioning novelty of the proposed approach, which suggests that the authors should pay special attention to that aspect when revising the paper.

Taking into account the initial reviews and rebuttal process, my opinion is that the paper should be accepted.